# TLR4 Deficiency Affects the Microbiome and Reduces Intestinal Dysfunctions and Inflammation in Chronic Alcohol-Fed Mice

**DOI:** 10.3390/ijms222312830

**Published:** 2021-11-27

**Authors:** Carlos M. Cuesta, María Pascual, Raúl Pérez-Moraga, Irene Rodríguez-Navarro, Francisco García-García, Juan R. Ureña-Peralta, Consuelo Guerri

**Affiliations:** 1Department of Molecular and Cellular Pathology of Alcohol, Prince Felipe Research Center, 46012 Valencia, Spain; cmcuesta@cipf.es (C.M.C.); maria.pascual@uv.es (M.P.); irant.12@gmail.com (I.R.-N.); 2Department of Physiology, School of Medicine and Dentistry, University of Valencia, 15 Avda. Blasco Ibanez, 46010 Valencia, Spain; 3Bioinformatics and Biostatistics Unit, Prince Felipe Research Center, 46012 Valencia, Spain; raulcl1994@gmail.com (R.P.-M.); fgarcia@cipf.es (F.G.-G.)

**Keywords:** TLR4, ethanol, intestinal microbiota, 16S rRNA, inflammation

## Abstract

Chronic alcohol abuse causes an inflammatory response in the intestinal tract with damage to the integrity of the mucosa and epithelium, as well as dysbiosis in the gut microbiome. However, the role of gut bacteria in ethanol effects and how these microorganisms interact with the immune system are not well understood. The aim of the present study was to evaluate if TLR4 alters the ethanol-induced intestinal inflammatory response, and whether the response of this receptor affects the gut microbiota profile. We analyzed the 16S rRNA sequence of the fecal samples from wild-type (WT) and TLR4-knockout (TLR4-KO) mice with and without ethanol intake for 3 months. The results demonstrated that chronic ethanol consumption reduces microbiota diversity and causes dysbiosis in WT mice. Likewise, ethanol upregulates several inflammatory genes (IL-1β, iNOS, TNF-α) and miRNAs (miR-155-5p, miR-146a-5p) and alters structural and permeability genes (INTL1, CDH1, CFTR) in the colon of WT mice. Our results further demonstrated that TLR4-KO mice exhibit a different microbiota that can protect against the ethanol-induced activation of the immune system and colon integrity dysfunctions. In short, our results reveal that TLR4 is a key factor for determining the gut microbiota, which can participate in dysbiosis and the inflammatory response induced by alcohol consumption.

## 1. Introduction

The gut microbiota is a highly complex ecosystem made up of millions of microorganisms that inhabit in the gastrointestinal tract. Most of these bacteria belong to the phyla Firmicutes (Gram-positive) and Bacteroidetes (Gram-negative) [1], and they produce numerous metabolites capable of influencing the host’s physiology. These organisms participate not only in digestion, but also in inflammation, immunity, and regulation of the gut–brain axis [2,3,4]. The combination of bacterial species is unique for everyone, but functions are highly preserved between individuals. Under healthy conditions, the microbiota is separated from the rest of the host by the intestinal mucosa barrier, which is formed by epithelial cells and by the proteins involved in mucosal homeostasis [5].

Microbial ecology stability is good; however, alterations to bacterial composition or activity can result in changes that may lead to an imbalance between pathogenic and beneficial microorganisms by triggering a process known as dysbiosis [6,7,8]. This pathological state is capable of inducing several diseases that impact the whole body, including brain cognitive and behavioral functions [9,10,11].

One factor that alters the microbiota equilibrium is ethanol consumption. Ethanol is able to induce bacterial overgrowth by modifying microbial composition in a harmful state [12], which leads to dysbiosis. Alcohol consumption may also promote the “leaky gut” disorder, characterized by increased intestinal mucosa that leads to the translocation of bacteria and their compounds through the intestinal membrane to blood circulation [13,14]. Many of these bacterial molecules have inflammatory properties, such as lipopolysaccharide (LPS), a structural component of the outer membrane of Gram-negative bacteria [15] that is recognized by Toll-like receptor 4 (TLR4). In fact, TLR4 responds less in the intestine than in other tissues [16] and maintains tolerance with commensal bacteria [17,18]. Higher plasma LPS levels have been detected in subjects with alcohol dependence that are able to induce deviations in host inflammatory responses [19].

We previously demonstrated that, by activating Toll-like receptor 4 (TLR4) in the brain and glial cells, ethanol can induce an immune response [20,21]. Activation of TLR4 signaling induces the release of cytokines and inflammatory mediators, and it is associated with inflammation in the gastrointestinal tract [18]. Indeed, alterations to either this receptor or its activity are linked with colitis [22,23]. Considering that chronic alcohol consumption alters the gut microbiota and intestinal barrier permeability, the present study assesses the potential role of the TLR4 response in not only intestinal bacterial diversity, but also in the dysfunctions associated with chronic ethanol intake. For this aim, we used wild-type (WT) and TLR4 knockout mice.

Our results support the notion that ethanol consumption modifies the bacterial profile and provokes gut inflammation in WT mice. We also demonstrated for the first time that TLR4-deficient animals show a distinctive intestinal microbiota that is more resistant to the inflammatory effects of ethanol. These results highlight the relevance of the TLR4 receptor in the development of the gut microbiota and intestinal injury induced by alcohol.

## 2. Results

### 2.1. Ethanol Treatment Upregulates Inflammation in the Intestinal Genes Associated with the TLR4 Response

Previous studies have demonstrated that ethanol treatment can damage the intestinal epithelium and cause dysbiosis [24]. Therefore, we evaluated the impact of alcohol treatment and the potential role of the TLR4 response in the microbiota community.

The first objective was to determine the potential inflammatory state of the gut from mice with and without ethanol treatment. To fulfill this objective, several messengers of the key inflammatory genes in colon tissues were measured by qPCR (Figure 1A). The RNA expression of *interleukin 1 beta* (*IL-1β*) and *inducible nitric oxide synthase* (*iNOS*) was significantly high in the alcohol-fed WT (WTE) mice compared to the control WT and ethanol-treated TLR4-KO mice (TLR4-KOE). Likewise, the *tumor necrosis factor alpha* (*TNF-α*) and *cyclooxygenase 2* (*COX-2*) levels were upregulated in the WTE mice group vs. the TLR4-KOE. The figure also shows that the *IL-10* levels in the WTE mice were upregulated in comparison to the WT group. Similarly, *CXCL10* (*C–X–C motif chemokine ligand 10*) showed an analogous pattern to the previous genes, with a higher WTE expression.

Next the miR 155-5p and miR 146a-5p levels were evaluated because our previous studies demonstrated that these miRNAs are induced in the cerebral cortex of ethanol-treated mice [25,26]. Figure 1B shows that both miRNAs were induced by chronic ethanol treatment in the WT mice. This also confirmed that the same ethanol treatment did not induce any significant change in the expression of these miRNAs in the TLR4-KO group. These results suggest that ethanol treatment triggers proinflammatory signal activity only in WT, but not in TLR-KO mice, which implies that microbiota composition differs in the WT and TLR4-KO mice genotypes.

However, compared to the WT control mice, neither the incorporation of alcohol into diet nor the different genotype seemed to alter food (2.48 ± 0.2 g/day in WT; 2.67 ± 0.17 g/day in chronic WT; 2.6 ± 0.25 g/day in TLR4-KO; 2.87 ± 0.34 g/day in chronic TLR4-KO) or liquid intake (3.07 ± 0.5 mL/day in WT; 2.86 ± 0.63 mL/day in chronic WT; 3.1 ± 0.55 mL/day in TLR4-KO; 3.67 ± 0.46 mL/day in chronic TLR4-KO) in animals. Figure 2 depicts solid and liquid intakes, as well as variations in mouse body weight.

### 2.2. Impact of TLR4 and Ethanol Consumption on the Microbiota Community

Then, 16S rRNA sequencing analysis of mice feces was performed to determine the actions of ethanol in the gut microbiome profile after 3 months of ethanol treatment. After cleaning and trimming, the final number of sequences used in the bioinformatic analysis averaged 19,221 per sample with a standard deviation of 15,877 (Appendix A). Next a Bray–Curtis-based nonmetric multidimensional scaling (NMDS) plot was employed to evaluate similarity between samples (PERMANOVA F-value: 3.1165, *R^2^*: 0.28035, *p*-value: 0.00099) (Figure 3A). Using these approaches, we assessed the structural similarities in microbiota communities in the WT vs. TLR4-KO and WTE vs. TLR4-KOE samples. Figure 3A shows a clear separation between WT and TLR4 (adjusted *p*-value < 0.01), which denotes the important role of TLR4 in bacterial composition. Additionally, the TLR4-KO and TLR4-KOE groups were separated (adjusted *p*-value < 0.01), while most of the WT and WTE samples were grouped into one cluster (adjusted *p*-value = 0.74).

The alpha diversity in mice feces was measured by two different indices: Chao1 and Shannon (Figure 3B). These alpha diversity indices indicated that the control groups WT and TLR4-KO had a higher average richness of bacterial species, and the degree of diversity was higher than in the ethanol-treated groups. These results show a lower bacterial diversity trend in the alcohol-treated animals compared to the control animals.

Notably, the bacterial diversity in the samples of the ethanol-treated groups, WTE and TLR4-KOE, showed more similarity than in the groups without ethanol. This low diversity could explain why the individuals belonging to the same ethanol group were similar according to NMDS (except for sample WTE-6) compared to the individuals in the untreated groups WT or TLR4-KO, which were more dispersed from one another.

### 2.3. TLR4 and Chronic Alcohol Treatment Transform the Gut Microbial in Adult Male Mice

Although we showed that alpha diversity was not significantly affected by ethanol, ethanol consumption and the TLR4 response modify the abundance of the taxonomic profiles in gut mice. In fact, by means of a differential analysis with the *DESeq2* package, the abundances of each taxonomic group and the significant differences in WTE vs. WT, TLR4-KOE vs. KO, and TLR4-KO vs. WT were evaluated.

Nine phyla were found, which predominated in most samples: Actinobacteriota, Bacteroidota (Bacteroidetes), Cyanobacteria, Deferribacterota, Desulfobacterota, Firmicutes, Patescibacteria, Proteobacteria, and Verrucomicrobiota (Figure 4A). Phyla Bacteroidetes and Firmicutes markedly dominated in all the groups. The abundance of these two phyla combined was around 90/95% of the total.

In relation to alcohol treatment, a similar decrease occurred in Gram-positive abundance in both comparisons (Figure 4B), WT vs. WE (42.75% vs. 31.62%) and TLR4-KO vs. TLR4-KOE (51.31% vs. 37.64%), although these reductions were not significant. Further studies are needed to confirm the differences found among groups in the WT vs. WTE comparison (statistical analysis of the comparisons are detailed in Appendix A).

On the abundance of bacteria’s phyla, Figure 5 shows how the Desulfobacterota phylum was reduced in two comparative groups, WTE vs. WT (*p* = 0.036) and TLR4-KOE vs. TLR4-KO (*p* = 0.022), which indicates the effect of ethanol treatment on this phylum. The Cyanobacteria phylum was also negatively affected by alcohol consumption in the TLR4-KO mice (*p* = 0.024). Interestingly however, the WT mice were not significantly affected by ethanol. The third phylum, Firmicutes, also significantly decreased in both KOE vs. KO (*p* < 0.001) and WT vs. KO comparisons (*p* < 0.001) and, as previously mentioned, was more dominant in the TLR4-KO mice. The last phylum, Actinobacteriota, was the only one in which ethanol intake led to population growth, with an increase in TLR4-KOE vs. TLR4-KO (*p* = 0.043), although no significant effect was observed when comparing groups WT vs. WTE. Table 1 and Appendix A summarize the obtained results.

### 2.4. Distinct Taxa Are Affected by Ethanol and the TLR4 Response in Feces Communities

In order to evaluate the potential alterations of bacteria abundance to other taxa, the *DESeq2* package was used to find ASVs (a higher-resolution and sensitive analog of the traditional OTU table) with statistically different expressions (adjusted *p*-value < 0.01) in three comparisons: WTE vs. WT, TLR4-KOE vs. TLR4-KO, and TLR4-KO vs. WT (Figure 6). All ASVs with an adjusted *p*-value < 0.01 are listed in Appendix A.

Ethanol impacted nine specific ASVs when comparing the WTE and WT mice. Figure 6A shows a large increment in the *Alloprevotella* genus and the Muribaculacetae family (Bacteroidota phylum) and a decrease in the different species of *Lachnoclostridium* and the *Lachnospiraceae_NK4A136_group* in the Lachnospiraceae family (Firmicutes phylum). However, in one of these genera (ASV61), ethanol treatment lowered their levels. Bacteria abundance in TLR4-KOE vs. TLR4-KO was then compared (Figure 6B). This comparison showed that 18 specific ASVs were statistically affected. Alcohol consumption increased the Erysipelotrichaceae family in two separate genera: *Dubosiella* and *Faecalibaculum*. In contrast, ethanol induced a negative effect on the Ruminococcaceae family, the *Rikenellaceae_RC9_gut_group* and the *Erysipelatoclostridium* genera, and on several genera of the Lachnospiraceae family (*Lachnospiraceae_UCG-006*, *Lachnoclostridium*, the *Lachnospiraceae_NK4A136_group*, and *Roseburia*, among others). It is interesting to note in the TLR4-KO mice that the significant changes in bacterial species induced by ethanol consumption were predominantly negative, with the biggest decrease in the *Roseburia* genus.

The genotype effect was evaluated last, by comparing TLR4-KO vs. WT. Figure 6C reveals a complex contrast between both genotypes, with 25 specific ASVs showing a statistically different expression. This suggests that these two genotypes displayed higher bacterial diversity.

The *DESeq2* package revealed that the absence of TLR4 brought about major alterations to gut microbiome composition, and opposite effects on the same genus were even observed when comparing WT and TLR4-KO (Figure 6). For instance, in both the Muribaculaceae family and the *Alloprevotella* genus, there were different ASVs with contrasting deregulations (±10–20 log_2_ fold change). The Ruminococcaceae family, the *Lachnospiraceae_NK4A136_group*, and *Papillibacter* genera were upregulated in the KO mice, conversely to the Muribaculaceae family. *Faecalibaculum*, *Dubosiella*, *Bifidobacterium*, and *Alistipes* genera were reduced in the absence of the TLR4 receptor. Notably, one relevant result was the marked reduction (–21.9 log_2_ fold change) in the *Bifidobacterium* genus, and the only ASV with a significant difference did not belong to the Firmicutes or the Bacteroidota phylum.

### 2.5. Role of TLR4 in Ethanol-Induced Alterations in Gut Structural Genes

By considering the involvement of TLR4 in inflammatory response and dysbiosis induced by ethanol in the colon epithelium, several structural genes associated with permeability were evaluated by analyzing the gene expression levels in the colon linked with different pathways (Figure 7A). For instance, the gene expression of intelectin 1 (ITLN1) was upregulated in the WTE group, but no significant effect was noted in chronic ethanol-treated TLR4-KO mice. The cystic fibrosis transmembrane conductance regulator (CFTR) gene (an epithelial ion channel) was substantially reduced in the WTE group, but the expression of this gene was higher in TLR4-KO than in the WT. Chronic ethanol intake also decreased the gene expression of cadherin 1 (CDH1) in both WT and TLR4-KO mice, but CDH1 expression was higher in TLR4-KO than in WT mice in both control and ethanol treatments.

### 2.6. Interactions between Inflammation and Integrity Genes and Gut Microbes

The possible relationship between the host genes and bacteria in all the groups was next investigated. For this purpose, the correlations between the structural genes with significant changes, and the three microbial taxa with different expressions, i.e., families Erysipelotrichaceae and Lachnospiraceae and phylum Firmicutes, were previously analyzed.

Using Spearman correlations, four significant (*p*-value < 0.1) gene–microbe correlations were observed (Figure 7B). While a negative correlation between microbes and genes was found in the groups with alcoholic treatment (*p*-value <0.1), no significant correlations appeared compared to the groups not treated with alcohol. Interestingly, the strongest negative correlation (*p*-value < 0.01) was observed between the Lachnospiraceae family and CFTR (Spearman’s rho = −1) in the WTE group. Other important negative correlations were ITLN1–Erysipelotrichaceae in KOE (Spearman’s rho = −0.89), CFTR–Firmicutes (Spearman’s rho = −0.94) in WTE, and CDH1–Lachnospiraceae (Spearman’s rho = −0.83) in WTE.

## 3. Discussion

Current evidence demonstrates the relevance of the gut microbiota on the host’s health for participating and interacting with other body tissues, and it is even sometimes considered to be an “essential organ” [27]. Indeed, the gut microbiome expresses approximately 150-fold more genes than the entire human genome [28]. Important advances have been made showing that the human gut microbiota not only provides a barrier to protect against foreign pathogens, but is also involved in nearly every human biological process [29]. However, alterations to the gut microbiota can reduce intestinal barrier integrity by leading to an increased leakage of lipopolysaccharides and fatty acids, which can act upon TLR4 to activate systemic inflammation [30]. Recent studies have also demonstrated that dysfunctions in the human microbiota participate in several diseases, such as inflammatory bowel disease (IBD), diabetes mellitus, metabolic syndrome, atherosclerosis, alcoholic liver disease, and neuroinflammation, and many of these diseases are associated with immune system dysfunctions [31,32]. Alcohol abuse also causes dysbiosis, intestinal inflammation, and epithelium permeability [13,33], and many of these disorders are associated with immune disturbances. We herein demonstrate that chronic alcohol consumption causes gut inflammation and microbiota alterations. These effects are linked with innate immune receptors, i.e., the TLR4 response, because alcohol slightly and differently affected the microbiome from the TLR4-KO mice. Our studies demonstrate for the first time that TLR4-KO mice exhibit a phenotype that is consistent with decreased inflammation.

TLRs are expressed in most epithelial cell lineages, and they recognize the different types of bacteria that participate in gut homeostasis. Tight regulation mechanisms prevent excessive responses toward commensal microorganisms [16,34]. Studies in alcoholic humans have demonstrated that chronic ethanol consumption changes the gut microbiome and induces dysbiosis [35,36]. Although the mechanisms that participate in the changes induced by alcohol and TLR4 are not completely understood, the present findings demonstrate that the TLR4-KO and the TLR4-KO alcohol-drinking mice showed a unique microbiota after 3 months of alcohol treatment, which differed from one another, as well as from their WT analogs. 

With the 16S rRNA gene sequencing analysis, various significant differential taxonomic shifts were observed between the alcohol-drinking mice and their respective controls, as well as between the WT mice and TLR4-KO mice. Long-term alcohol consumption has been generally shown to increase Gram-negative bacteria growth [37]. The present work observed an increase in gut Gram-negative bacteria in both WT and TLR4-KO mice, but these effects were not significant. Endotoxin LPS is present in the outer membrane of most Gram-negative bacteria [38] and can cause intestinal inflammation [39]. In fact, a microbiota dominated by Gram-negative bacteria, which may be related to a higher LPS concentration, results in greater predisposition to generate a pathology associated with dysbiosis [40,41]. However, we observed that bacterial Gram-positive taxa declined with alcohol intake. Unlike other studies [42], our results showed that chronic alcohol treatment decreased the phylum Firmicutes, the dominant Gram-positive phylum in the intestine, which is possibly due to the reduction in the Lachnospiraceae family. This family is dominantly expressed in the gut of healthy adult human individuals [43], but is strongly affected by alterations to the bacterial community [44]. Furthermore, all Lachnospiraceae members are anaerobic, and ethanol metabolism mediates ROS production [45], causing the enrichment of aerotolerant species that are more resistant to this oxidative environment [46]. Both the Muribaculaceae family and the *Alloprevotella* genus significantly increased in the WT mice with alcohol drinking, which corroborates previous findings in alcoholic patients [47]. Lastly, the *Alloprevotella* genus has been associated with intestinal permeability and inflammation [48,49], which are common effects of alcohol consumption. 

It is notable that the TLR4-KO mice treated with alcohol exhibited different changes in their microbiota compared to the WT animals. For instance, the Erysipelotrichaceae family increased in the TLR4-KO mice treated with alcohol, but no significant changes took place in the WT mice treated with alcohol, as reported in other studies [50,51]. Our study also observed that Erysipelotrichaceae genera increased in chronic TLR4-KO mice, specifically, *Dubosiella* and *Faecalibaculum*. These two genera have been recently discovered and studied in the murine intestine [52]. Hence, their interactions with the host’s TLR4 and with ethanol need to be characterized. In addition, the genera that was most negatively affected in the TLR4-KO and WT mice with alcohol intake were *Roseburia* and *Lachnoclostridium,* respectively. Both groups have been shown to decrease with alcohol consumption in humans [53,54] and are butyrate-producing microorganisms [55,56]. Studies in humans have demonstrated that alcohol consumption reduces the butyrate concentration, a short-chain fatty acid (SCFA) with many health benefits in the intestine compared to individuals without alcohol consumption [57]. The reduction in butyrate-producing species in alcoholic individuals would explain the lower butyrate concentration in their intestine.

Intestinal microbiota composition is conditioned to the individual immune system [58] and the TLR4 response [34,59]. We provide evidence that the microbiota of both WT and TLR4 mice undergoes distinct changes in the presence of ethanol. Although our studies suggest a connection between alcohol-mediated inflammation and toxicity in mice and their microbiota, the direct role of alcohol in microorganisms as calorie intake cannot be ruled out.

LPS is an important ligand of TLR4 [60], which might be related to the lowering Gram-negative ratio in the TLR4-KO mice. Indeed, microbiome-derived LPS can facilitate host tolerance of gut microbes in a mechanism in which TLR4 participates [61,62]. Therefore, the absence of TLR4 can adversely affect the coexistence of Gram-negative bacteria with a host’s immune system. For instance, the genus *Alistipes*, a potential opportunistic pathogen [63] that produces LPS with low immunostimulatory capacity [61], is reduced in TLR4-KO mice. The authors of [64] observed that *Alloprevotella* and *Bacteroides* populations decreased in the TLR4-KO group, and both were Gram (–). In our study, some species from the *Alloprevotella* genus decreased in the TLR4-KO animals, but expanded in others. Another genus that expanded in the TLR4-KO mice, *Papillibacter*, has been shown to decrease in patients with Parkinson’s disease with increased TLR4 expression [65]. The *Bifidobacterium* genus is a bacterium with a marked capacity to modulate the TLR4 pathway by inhibiting its expression [66]. Therefore, the absence of an immune innate receptor such as in TLR4-KO mice, to which *Bifidobacterium* is adapted, could negatively affect this bacterium.

Studies have also demonstrated that alcohol abuse can cause intestinal inflammation by increasing intestinal epithelium permeability, as well as by affecting intestinal immune homeostasis [33,67,68,69]. In fact, the microbiota also participates in alcohol-induced gut inflammation. For instance, many facultative anaerobes and aerobes bacteria exhibit catalase activity and can metabolize ethanol to acetaldehyde, which promotes inflammatory effects [37,70,71]. Accordingly, the present findings further demonstrated that chronic alcohol consumption upregulated colon mRNAs (IL-1β, iNOS, TNF-α, COX-2, IL 10, and CXCL10) and miRNAs (miR 155-5p and miR 146a-5p), associated with inflammatory effects in the WT mice, but not in the TLR4-KO animals with the same ethanol treatment. Our results showed that TLR4-KO mice were protected from the effect of ethanol in gut inflammation. This effect is probably not only due to the absence of the TLR4 receptor, but also because of the different microbiota when this receptor is lacking. Indeed, microbiota have also been involved in anti-inflammatory effects [72,73]. The lower abundance of Gram-negative bacteria in chronic ethanol-treated TLR4-KO mice vs. chronic WT mice suggests lower concentrations of LPS and its inflammatory activity.

Ethanol also has the capability to affect the structure and channels of the intestinal epithelium and leads to a significant increase in membrane permeability and the translocation of potentially toxic metabolites [67]. This is evidence to demonstrate that ethanol brings about a progressive disruption of barrier proteins [74,75,76]. Several studies have shown that an increase in intestine epithelium permeability and endotoxin levels in plasma correlates positively with the amount of ingested alcohol [75,76].

The present findings demonstrated that chronic alcohol abuse modified the expression of genes ITLN1 and CFTR in WT mice intestines with minor changes in TLR4-KO compared to the untreated control mice. Another gene with changes was CDH1, a cadherin involved in cell–cell adhesions and cell invasion suppression that makes a prominent contribution to epithelial maintenance during inflammation [77]. This gene is downregulated when exposed to ethanol [78] and in inflamed areas of the intestine [79]. Our results confirmed that, although alcohol abuse can induce colonic inflammation and affect intestinal functions and microbiota composition, the damaging effects of ethanol were more prominent in the WT than in the TLR4-KO mice, probably due to the lack of an innate immune response associated with TLR4 receptors, as well as to the development of the different microbiota.

Several studies have shown that specific bacteria are associated with intestinal membrane integrity. For instance, the *Lachnospiraceae_NK4A136_group* genus is an important SCFA-producing and homeostatic bacterium [80,81] that was more abundant in the TLR4-KO control mice than in the WT control mice. Several of these taxa, such as the *Lachnospiraceae_NK4A136_group* genus, remained more abundant in TLR4-KO compared to the WT mice, even when alcohol treatment exerted a hypothetical anti-inflammatory effect. Actinobacteriota is an important phylum in gut homeostasis development and maintenance [82]. We observed how this phylum increased in the ethanol-treated TLR4-KO mice, whereas phylum Firmicutes decreased in these animals. These results suggest that the vacant niche left by Firmicutes in this alcoholic gut ecosystem can be replaced with other nonpathogenic species such as Actinobacteriota instead of other more immunogenic Gram-negative species.

## 4. Materials and Methods

### 4.1. Animal Model

Male WT (TLR4^+/+^) (Harlan Ibérica S.L., Barcelona, Spain) and TLR4 knockout (TLR4-KO, TLR4^−/−^, KO) mice, kindly provided by Dr S. Akira (Osaka University, Osaka, Japan) with C57BL/6 genetic backgrounds, were used. Animals were kept under controlled light and dark conditions (12 h/12 h) at a temperature of 23 °C and 60% humidity. Animal experiments were carried out in accordance with the guidelines set out in European Community Council Directive (2010/63/ECC) and Spanish Royal Decree 53/2013 modified by Spanish Royal Decree 1386/2018 and were approved by the Ethical Committee of Animal Experimentation of the CIPF (Valencia, Spain).

### 4.2. Alcohol Treatment

Twenty-eight 2 month old animals were used for this work. They were housed (2–4 animals/cage) and divided into four groups, eight WT C57BL/6J mice and eight TLR4-KO mice for the control conditions, along with six WT C57BL/6J and six TLR4-KO for the alcohol chronic treatment. Mice were treated with water (WT and KO) or water containing 10% (*v*/*v*) alcohol (WTE and KOE) for 3 months. They ate a solid diet ad libitum for these 3 months (48% carbohydrate, 14.3% protein, and 4% fat; 2.9 kcal/g energy density). Food was the 2014 Teklad global 14% protein rodent maintenance diet (Envigo, Sant Feliu de Codines, Barcelona, Spain). Daily food and fluid intake in the four groups was carefully measured for 3 months. No health problems were shown. From previous studies that we carried out with the same treatment, we know that blood ethanol levels are comparable in the treated groups (WTE and KOE) (125 ± 20 mg/dL) [83,84]. lastly, body weight gain at the end of the 3 month period was similar in all the groups. The experiment design is illustrated in Figure 8.

### 4.3. Sample Collection

Fecal samples were collected from the four groups (WT, WTE, KO, and KOE) and stored at the end of treatment with or without ethanol (at 5 months of age). The samples from all the mice in the different groups were collected fresh and individualized. At the end of alcohol treatment, stool pellets were collected, and mice were euthanized by CO_2_. Tissues were collected from the cortex and colon, immediately frozen in liquid nitrogen, and stored at −80°C until used for further analyses.

### 4.4. Bacterial 16S rRNA Library Construction and Sequencing 

DNA was extracted using the QIAamp Fast DNA Stool Mini Kit (Qiagen, Hilden, Germany) according to the manufacturer’s protocol with the following modifications: the stool and tissue samples were homogenized in 100 µL of InhibitEx buffer for 5 min at 95 °C. To elute DNA, 100 µL of Buffer ATE was used. The isolated DNA quality control was measured by a spectrophotometer (NanoDrop 2000; Thermo Scientific, Waltham, MA, USA). The extracted bacterial DNA was stored at −20 °C until used for PCR amplification and sequencing purposes. 

The prokaryotic 16S ribosomal RNA gene (16S rRNA) libraries were generated from the extracted DNA using amplicon PCR for the variable V3–V4 region and the Illumina 16S Sample Preparation Guide. First, the region of interest was amplified using the specific primers with attached overhang adapters (Table 2). Unbound primers, primer-dimer fragments, and other contaminants were removed using AMPure XP beads (Illumina, San Diego, CA, USA) according to the above guide. A second PCR was performed using Nextera XT Index 1 Primers (N7XX) and 2 Primers (S5XX) (Illumina). Appendix A shows the employed indices for the Sensitivity DNA 1000 assay (Agilent Technologies, Waldbronn, Germany).

The concentration was evaluated using the Qubit^®^ dsDNA BR Assay kit. Using the amplicon size and Qubit concentrations, samples were normalized to 10 nM. A pool of amplicons at 10 nM was created by adding 2.5 µL of each normalized amplicon to a single pool. The pool was re-quantified by a TapeStation High Sensitivity DNA 1000 assay (Agilent Technologies, Waldbronn, Germany) and Qubit. Sequencing was performed on the Illumina MiSeq platform (Illumina) following a 250 bp paired-end protocol according to the manufacturer’s specifications with the addition of 25% PhiX. The library concentration was optimized to 8 pM. The experimental design is shown in Figure 8A.

### 4.5. 16S rRNA Gene Sequencing Analysis

The *DADA2 R* package was used to carry out the quality control of readings and alignments. This pipeline filters the quality of readings by considering the amount of maximum expected errors and the number of *N* in the sequences, as well as trims the sequences that have passed the filter. *FastQC* (0.11.9) and *MultiQC* (1.8) were employed to obtain a comprehensive vision of the sequencing quality in the overall sample set. The Divisive Amplicon Denoising Algorithm (DADA) was applied. Read merging was performed when there was at least 12 bp that overlapped between both reads and reads that were eliminated chimeras. The product was an amplicon sequence variant (ASV) table, a higher-resolution and sensible analog of the traditional OTU table [85]. Plots were generated with the ggplot package. The data generated in this work are available in the SRA public repository, with the following accession ID: PRJNA780249.

Sequences were classified from the phylum down to the species level. Exact matching against a reference sequence was required to assign the species level. Silva version 138 was used as a reference database to identify the taxonomic classification of sequence reads. This was achieved with the functions implemented in the DADA2 package: *assingSpecies* and *assingTaxonomy*. The default arguments of functions were employed.

The samples herein used with their size molarity after extraction, total reads, final viable readings, and the taxonomized percentage of each taxon are summarized in Appendix A. The ASVs from the same phylum or Gram classification were merged to their respective group/classification. Permutational multivariate analysis of variance (PERMANOVA) tests for ASVs matrix with Bray–Curtis distance were computed using the vegan function Adonis. Differences among groups were computed with pairwise PERMANOVAs. We utilized the pairwise Adonis function, and *p*-values were corrected by the Bonferroni method.

### 4.6. Total RNA Isolation, Quantity, and Quality Determinations

The colon tissue samples were employed for total RNA extraction using Trizol followed by the phenol chloroform method and stored at −80 °C. RNA was measured using a NanoDrop ND-1000 Spectrophotometer (260/280 nm ratio) for the control quality.

### 4.7. Reverse Transcription and Real-Time Quantitative PCR for Total RNA and miRNA

For mRNA, 1000 ng of total RNA was retrotranscribed with the NZY First-Strand cDNA Synthesis Flexible Pack (NZYtech) following the manufacturer’s protocol. For miRNA, 100 ng of total RNA from the colon tissue was retrotranscribed using the TaqMan™ Advanced miRNA cDNA Synthesis Kit (Applied Biosystems, Barcelona, Spain) according to the manufacturer’s protocol. The reaction was carried out in a Master cycler ep. 5341 (Eppendorf AG, Hamburg, Germany). cDNA was stored at −20 ’C. RT-qPCR was performed in a LightCycler^®^ 480 System (Roche, Mannheim, Germany). The sequences of the forward and reverse primers for mRNAs, as well as the primers and chromosome location for miRNAs, are listed in Table 2. The relative expression ratio of a reference/target gene was calculated according to the Pfaffl equation. Housekeeping cyclophilin-A (Ppia) was used as a reference gene for mRNAs and snRNA U6 for miRNAs.

### 4.8. Statistics

Custom R scripts were applied in the bioinformatics analysis. A statistical and bioinformatics analysis was carried out with R, Version 3.6.3 (R Core Team, R Foundation for Statistical Computing, Vienna, Austria). An exploratory analysis of the metagenomic data was run in a reduced dimensional space using Bray–Curtis-based nonmetric multidimensional scaling (NMDS). To do so, the ASV count was transformed into proportions using the function *transform_sample_counts* and ordinates to compute NMDS. This function belongs to the *phyloseq* (1.30.0) package.

Alpha diversity was calculated to estimate both microbiota diversity and abundance. The Shannon index was used to measure species evenness (similar abundance level between species), and the Chao1 index was used to estimate microbiome richness or abundance in samples. A Mann–Whitney U test was performed to test if there were any significant differences between groups. The function *estimate_richness* was applied to compute alpha diversity and the Chao1 index from the *phyloseq* package. Lastly, the function *pairwise.Wilcox.test* from the stats, 3.6.2 version (R Core Team, R Foundation for Statistical Computing) package was used to compute a Mann–Whitney U test to check if there were any significant differences between groups. The *DESeq2*, version 1.26.0 (Heidelberg, Germany) package was employed for differential abundance testing. The library size was corrected by the *poscount* argument in the *sizefactor* function. Multiple-inference correction of *p*-values was performed following the Benjamini–Hochberg method.

The Gram-positive/negative ratio differences between groups were analyzed using the Kruskal–Wallis test following Dunn’s multiple comparison test corrected by the Sidak method. Significant differences in the gene expression between groups were calculated using the GraphPad Prism 8 Software (San Diego, CA, USA) through a one-way ANOVA followed by Tukey’s multiple comparison test. The results were expressed as the mean ± SEM. A correlation analysis was performed using Spearman’s correlation, and centered log-ratio transformation was applied to the ASV table before computing the correlation. Visualization and plotting were carried out with the ggplot (3.3.2) package. The employed bioinformatic pipeline is also detailed in Figure 8B.

## 5. Conclusions

In short, our results confirm that ethanol causes an inflammatory response in the gut tract by changing intestinal mucosa and the epithelium integrity, as well as by causing dysbiosis in the gut microbiota. We also demonstrate, for the first time, that lack of immune receptor TLR4 promotes a different microbiota with lower Gram-negative bacteria abundance, which could help to protect against ethanol-induced inflammation. These findings can open up new avenues to understand the relationship between the involvement of TLR4 and the gut microbiota in ethanol-mediated inflammation in the digestive system.

## Figures and Tables

**Figure 1 ijms-22-12830-f001:**
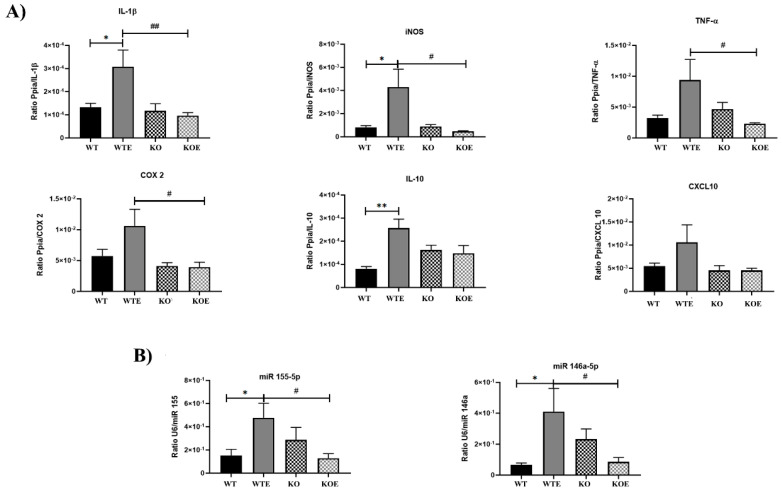
RT-qPCR differential expression analysis. The graphics represent the RT-qPCR comparisons made from the colons of the WT and TLR4-KO mice, with and without ethanol treatment. (**A**) Colon expression of the proinflammatory *interleukin 1β*, enzyme *iNOS*, cytokine *TNF-α*, enzyme *COX-2*, *interleukin 10*, and chemokine *CXCL10* mRNA levels. (**B**) Colon expression of the miR-155-5p and miR-146a-5p levels. * *p* < 0.05, ** *p* < 0.01 for treatment comparison purposes, and ^#^ *p* < 0.05, ^##^ *p* < 0.01 for the genotype comparison according to one-way ANOVA and Tukey’s multiple comparisons test, except for CXCL10, according to the Kruskal–Wallis test and Dunn’s comparisons test. *N* = 6. Bars represent the mean ± SEM.

**Figure 2 ijms-22-12830-f002:**
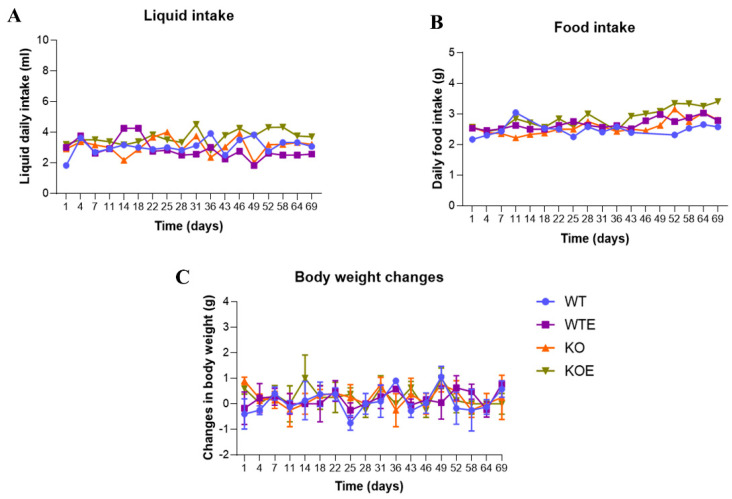
Diet dynamics in mice. (**A**) Liquid intake (water and water containing 10% (*v*/*v*) alcohol) in the WT, chronic ethanol-treated WT, TLR4-KO, and chronic ethanol-treated TLR4-KO animals during treatment. (**B**) Solid food intake in the four experimental groups of animals during treatment. (**C**) Variations in mouse body weight throughout alcohol treatment in the four experimental groups. No significant changes were observed in the different experimental groups. *N* = 4 per group.

**Figure 3 ijms-22-12830-f003:**
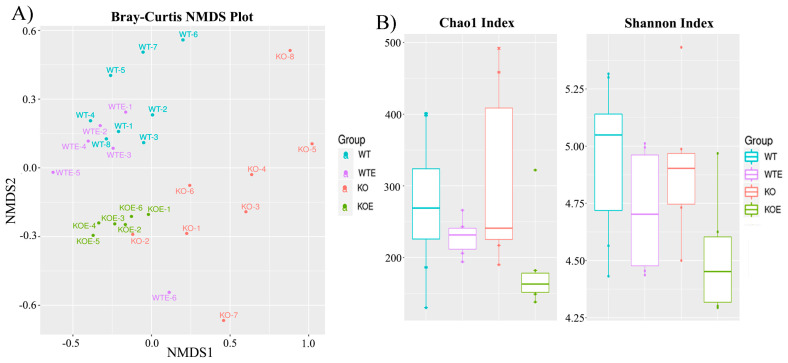
Bray–Curtis NMDS plot of fecal samples, showing that ethanol treatment results in lower species richness and phylogenetic diversity. (**A**) The fecal microbiota of the four groups showing three differentiated clusters. (**B**) Bacteria diversity was analyzed using the Chao1 and Shannon indices. Microbiota alpha-diversity was lowered by ethanol consumption independently of the genotype, and lack of TLR4 had no effect on the number of species. KO vs. KOE were more different groups (*p* = 0.076 and *p* = 0.13 in the Chao1 index and the Shannon index, respectively), whereas WT vs. KO were more similar groups (*p* = 0.959 and *p* = 0.51 in the Chao1 index and the Shannon index, respectively). Statistics obtained according to the Mann–Whitney U test with *p*-values corrected by Benjamini and Hochberg, *N* = 6.

**Figure 4 ijms-22-12830-f004:**
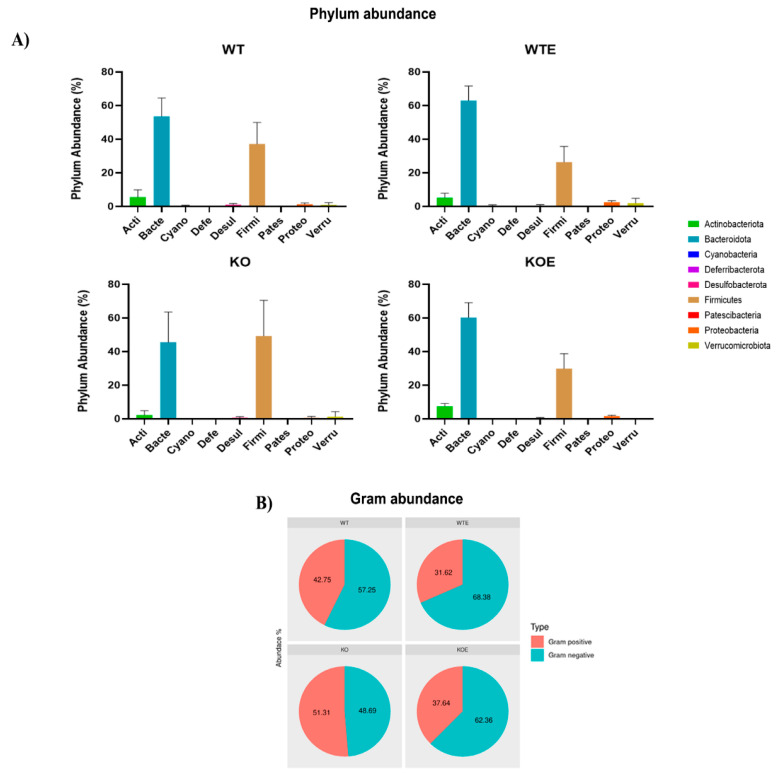
Microbiota composition of each group and changes among them. (**A**) Average phylum abundance in each group, where the dominance of phyla Firmicutes and Bacteroidota phyla is seen. (**B**) Pie charts depicting the average Gram-positive and Gram-negative abundance in the four groups.

**Figure 5 ijms-22-12830-f005:**
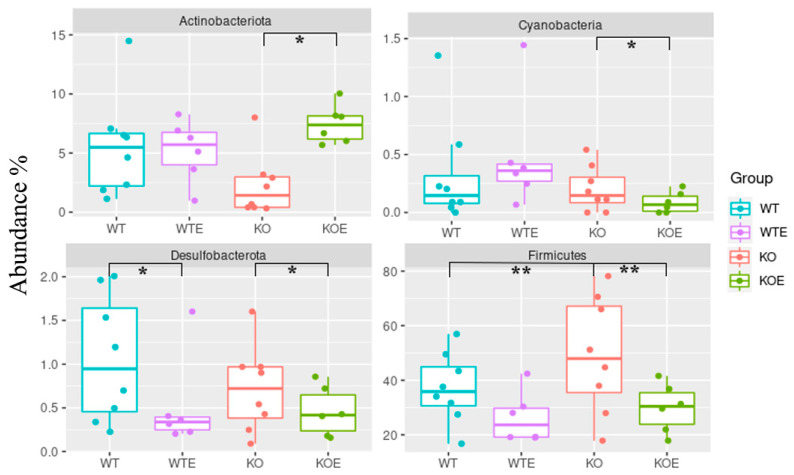
Influence of alcohol and genotype on mice in the four dominant phyla. The abundances of each phylum were assessed for significant differences in WTE vs. WT, KOE vs. KO, and KO vs. WT. A differential expression analysis was determined using the DESeq2 package and is shown in Table 1. Library normalization was performed using the poscounts method and local fitting for the negative binomial distribution model. The Benjamini–Hochberg method was applied to perform *p*-value correction. * *p* < 0.05, ** *p* < 0.01.

**Figure 6 ijms-22-12830-f006:**
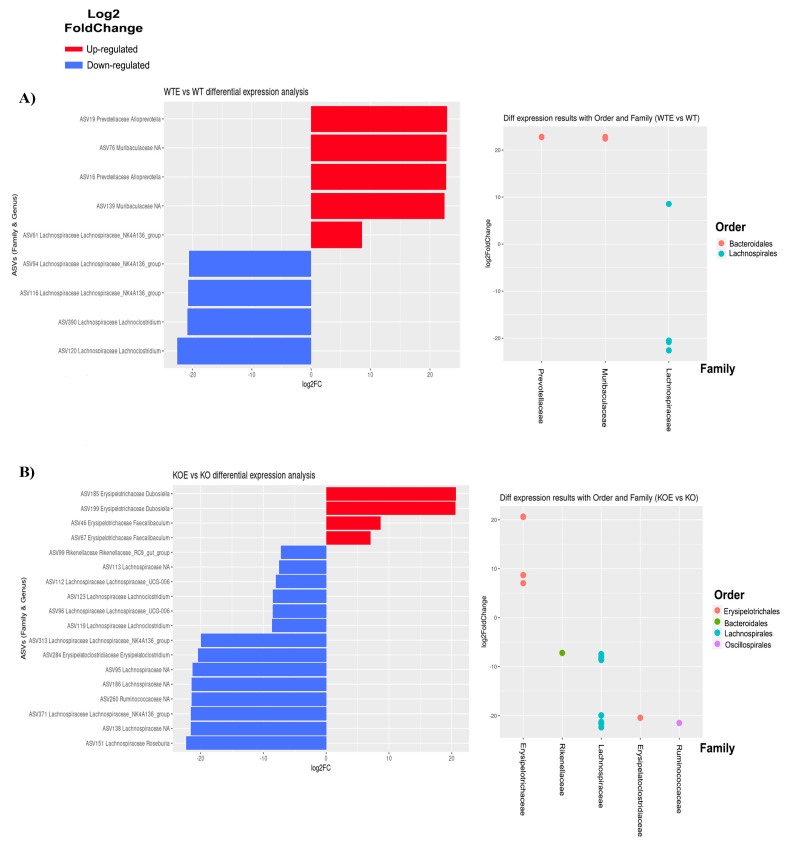
Differential expression analysis in bacteria order, family, and genus among groups. Log_2_ fold change bars with statistically different expressed ASVs among groups. Each ASV with one species, nine in WTE vs. WT, 18 in KO vs. KOE, and 25 in KO vs. WT. Red and the right position indicate enriched species, and blue and the left position denote depleted species. (**A**) WTE vs. WT, (**B**) KOE vs. KO, and (**C**) KO vs. WT order, family, and genus comparisons. Tests for the differential expression were done using the DESeq2 package. Only the ASVs with an adjusted *p*-value < 0.01 are shown.

**Figure 7 ijms-22-12830-f007:**
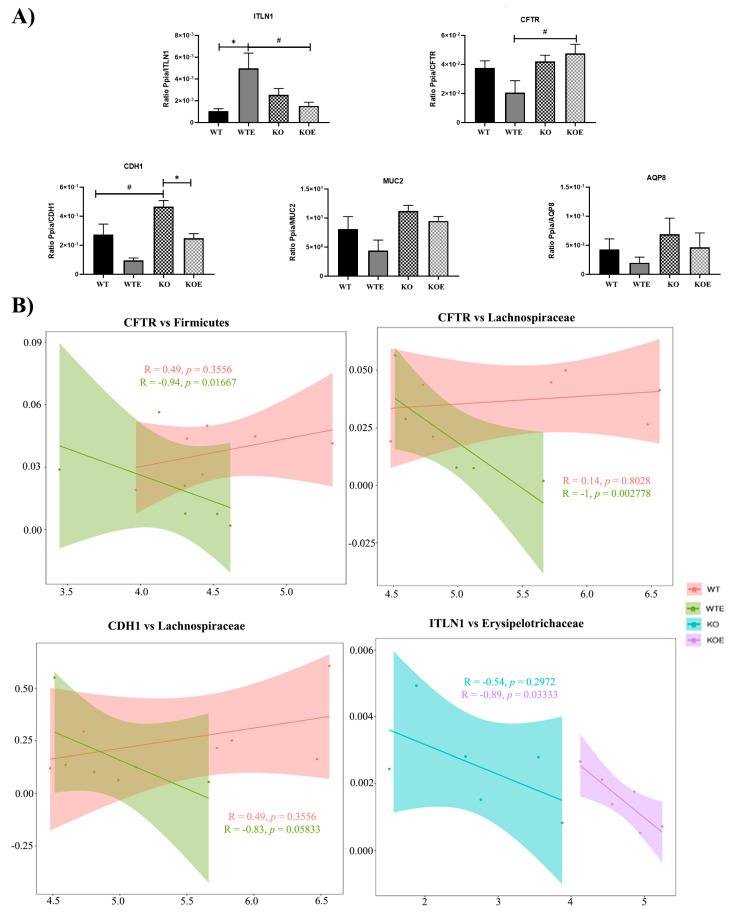
RT-qPCR differential expression analysis and gene–bacteria correlations. (**A**) The graphics represent the RT-qPCR comparisons made from the colons of the WT and TLR4-KO mice, with and without ethanol treatment. Colon expression of the lectin ITNL1, protein CFTR, cadherin CDH1, mucin 2, and aquaporin 8 mRNA levels. * *p* < 0.05 for treatment comparison, and ^#^ *p* < 0.05 for genotype comparison according to one-way ANOVA and Tukey’s multiple comparisons test, except for MUC2 and AQP8, according to the Kruskal–Wallis test and Dunn’s comparisons test. *N* = 6. Bars represent the mean ± SEM. (**B**) Scatterplots depicting the correlation analysis between the genes associated with intestine health and gut microbes with differential abundance within groups. The strength of the correlation (Spearman’s rank correlation coefficient) is represented as *R*) and significance is represented as *p*. Both are indicated in each plot.

**Figure 8 ijms-22-12830-f008:**
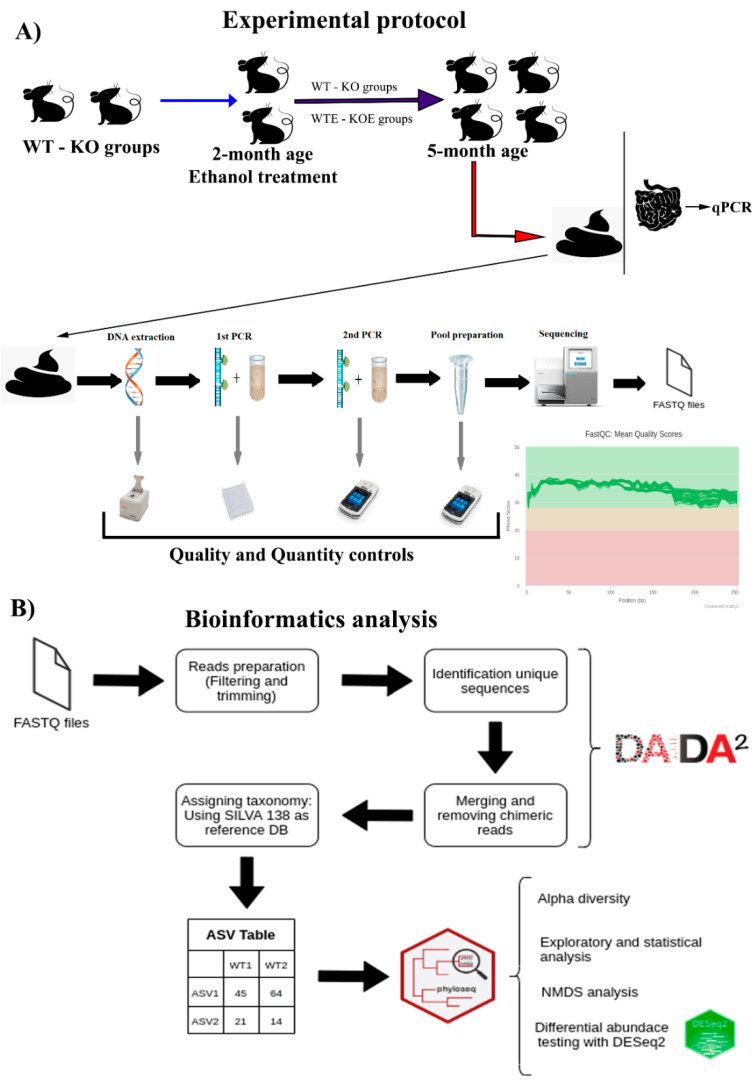
Schematic workflow of the experimental design and analysis. (**A**) Male mice with a CC57BL/6J background, with and without the deletion of TLR4, were either subjected to water containing 10% (*v*/*v*) alcohol at the 2 month time point or maintained as parallel untreated controls. Stool and colon tissues were collected from all the mice at 5 months. DNA was extracted from feces, and genomic 16S rRNA was sequenced. (**B**) ASVs were aligned to reference databases and analyzed for alpha diversity, NMDS, differential abundance and statistical analysis.

**Table 1 ijms-22-12830-t001:** Differential expression in the bacteria phylum in WTE vs. WT, KOE vs. KO, and KO vs. WT. Reads of each ASV were merged to their respective phylum level, and a differential expression was conducted using the DESeq2 package. Only the phyla with a raw *p*-value <0.05 are shown.

	baseMean	Log_2_ Fold Change	lfcSE	Stat	*p*-Value	*p*-adj	Kingdom	Phylum
**WTE vs. WT**								
ASV114	114.434	−1.270113	0.60419	−2.1022	0.03554	0.31985	Bacteria	Desulfobacterota
**KOE vs. KO**								
ASV5	751.696	1.292251	0.63923	2.0216	0.04322	0.09724	Bacteria	Actinobacteriota
ASV13	6441.341	−1.388448	0.35495	−3.9117	0.00009	0.00082	Bacteria	Firmicutes
ASV114	114.434	−1.384593	0.60594	−2.2850	0.02231	0.07158	Bacteria	Desulfobacterota
ASV336	42.115	−1.928905	0.85374	−2.2594	0.02386	0.07158	Bacteria	Cyanobacteria
**KO vs. WT**								
ASV13	6441.341	1.197066	0.32857	3.6432	0.00027	0.00242	Bacteria	Firmicutes

Bold words indicate the comparative.

**Table 2 ijms-22-12830-t002:** Primer sequences for the targeted mouse genes and miRNAs (RT-PCR assay).

**Genes**	**Forward (5′–3′)**	**Reverse (5′–3′)**
16S rRNA V3–V4 region	TCG TCG GCA GCG TCA GAT GTG TAT AAGAGA CAG CCT ACG GGN GGC WGC AG	GTC TCG TGG GCT CGG AGA TGT GTA TAA GAGACA GGA CTA CHV GGG TAT CTA ATC C
snRNA U6	GCT TCG GCA GCA CAT ATA CTA AAA T	CGC TTC ACG AAT TTG CGT GTC AT
Ppia	CGC GTC TCC TTC GAG CTG TTT G	TGT AAA GTC ACC ACC CTG GCA CA
IL-1B	CTC ATT GTG GCT GTG GAG AA	TCT AAT GGG AAC GTC ACA CA
iNOS	AAT CTT GGA GCG AGT TGT GG	AAT CTC TGC CTA TCC GTC TCG
TNF-alfa	GAA CTG GCA GAA GAG GCA CT	AGG GTC TGG GCC ATA GAA CT
COX 2	CAT TGA CCA GAG CAG AGA GAT G	GGC TTC CAG TAT TGA GGA GAA C
IL-10	AGG CGC TGT CAT CGA TTT CT	ATG GCC TTG TAG ACA CCT TGG
CXCL10	ATG ACG GGC CAG TGA GAA TG	TCA ACA CGT GGG CAG GAT AG
ITNL1	TCC AGT CAG CAA GGC AAC AGA G	CAG GTT CTC AGC CTG GAT GTC A
CFTR	CCA TCA GCA AGC TGA AAG CAG G	GTA GGG TTG TAA TGC CGA GAC G
CDH1	GGT CAT CAG TGT GCT CAC CTC T	GCT GTT GTG CTC AAG CCT TCA C
Muc2	GCT GAC GAG TGG TTG GTG AAT G	GAT GAG GTG GCA GAC AGG AGA C
AQP8	AAC TTG TGG GCT CCG CTC TCT T	ACA GCA GGG TTG AAG TGT CCA C
**miRNA**	**Chromosome Location**	**Sequence**
hsa-miR-146a-5p	Chr.5: 160485352 to 160485450 [+] on Build GRCh38	UGAGAACUGAAUUCCAUGGGUU
mmu-miR-155-5p	Chr.16: 84714140 to 84714204 [+] on Build GRCm38	UUAAUGCUAAUUGUGAUAGGGGU

## Data Availability

The data presented in this publication were deposited in the SRA public repository and are accessible through the following accession ID: PRJNA780249 (https://www.ncbi.nlm.nih.gov/Traces/study/?acc=PRJNA780249&o=acc_s%3Aa).

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
