# Peer review of "TLR4 Deficiency Affects the Microbiome and Reduces Intestinal Dysfunctions and Inflammation in Chronic Alcohol-Fed Mice"

_ijms, 2021, doi:10.3390/ijms222312830_

Round 1

Reviewer 1 Report

TLR4 is thought to play significant roles in the interaction between gut microbiota and hosts. This study was conducted on the relationship between alcohol, gut microbiota, TLR4, and intestinal immunity.

I think this topic is a very important topic in the effect of gut microbiota on the human health. Nevertheless, this study showed each correlations, but did not prove causal relationships.

Major

1. The main hypothesis of this study is unclear. What relationship between alcohol, TLR4, gut microbiota, and gut immunity is the main hypothesis of this study?

2. There is no proof of causal relationships and mechanisms.

To assert that the changes in the gut microbiota caused by TLR4 K/O prevented the inflammatory effects of alcohol, it is necessary to either see the effect of only the changes in the gut microbiota through fecal microbiota transplantation or to elucidate the mechanism by which changed bacteria reduce inflammation. This study also cannot prove that TLR4 acts on the mechanism by which alcohol changes gut microbiota.

3. Repeat experiments are necessary.

Minor

1. Introduction: Description about the Gut-brain axis seems unnecessary. Rather, it seems that the relationship between TLR and gut bacteria and the relationship between TLR and gut inflammation need to be discussed more.

2. 4 pages 123~127

In order to assert for differences between groups, statistical differences through Bray-Curtis distance should be shown.

3. 5 page 136~140

Table 1. and Shannon (Figure 3B).: This sentence is not understood.

Is it necessary to express the P-value between WE and KOE in this sentence?

There was a trend, but no statistical difference was observed.

4. Figure 6: It is necessary to increase readability by expressing the dominant group according to the direction.

Author Response

Major comments

1-The main hypothesis of this study is unclear. What relationship between alcohol, TLR4, gut microbiota, and gut immunity is the main hypothesis of this study?

Thank you very much for your comment. In fact, we have now altered the main hypothesis of this study as, “The present study aims to assess the potential role of TLR4 response in the intestinal bacteria diversity and the dysfunctions associated with chronic ethanol intake. For this aim we used wild-type and TLR4 knockout mice”, line 60.

2.- There is no proof of causal relationships and mechanisms.

To assert that the changes in the gut microbiota caused by TLR4 K/O prevented the inflammatory effects of alcohol, it is necessary to either see the effect of only the changes in the gut microbiota through fecal microbiota transplantation or to elucidate the mechanism by which changed bacteria reduce inflammation. This study also cannot prove that TLR4 acts on the mechanism by which alcohol changes gut microbiota.

We agree that further experiments are needed to clarify the influence of TLR4-KO in the gut microbiota and in the inflammatory effects of alcohol. However, our results demonstrated some important points that support this statement. For instance:

In line 329: “A lower abundance of bacteria Gram (-) in chronic ethanol TLR4-KO mice vs the chronic WT mice suggest lower concentrations of LPS and its inflammatory activity”.

In line 347: “Several of these taxa, such as Lachnospiraceae_NK4A136_group genus, remain more abundant in TLR4-KO compared to WT even when alcohol treatment, exerting a hypothetical anti-inflammatory effect”.

In line 471: “as a lower bacteria Gram (-) abundance”.

 3.- Repeat experiments are necessary.

We agree that further experiments are needed to confirm the present results. However, the present data support that elimination of TLR4 response (TLR4-KO) exerts some beneficial effects in the intestinal microbiota, since TLR4-KO mice exhibit a different microbiota that can protect against ethanol-induced activation of the immune system and colon integrity dysfunctions.

Minor comments

-1. Introduction: Description about the Gut-brain axis seems unnecessary. Rather, it seems that the relationship between TLRs and gut bacteria and the relationship between TLRs and gut inflammation need to be discussed more.

As suggested, we have further discussed the relationship between TLRs and gut bacteria and TLRs and gut inflammation.

We have added in line 52 “which is recognized by the toll-like receptor 4 (TLR4). In fact, TLR4 has a lower response in the intestine than in other tissues, maintaining the tolerance with the commensal bacteria”. Line 58 “mediators, and is associated with inflammation in the gastrointestinal tract. In fact, alterations in this receptor or in its activity are linked to colitis”.

We have also removed references associated with the nervous system that were not relevant in the present work following your advice.

 -2. 4 pages 123-127, In order to assert for differences between groups, statistical differences through Bray-Curtis distance should be shown.

As suggested, differences between groups and statistical differences using Bray-Curtis are now included in line 113, “(PERMANOVA F-value: 3.1165, R-squared: 0.28035, p-value: 0.00099)”, and in the same paragraph, the p-values obtained in the comparatives.

-3. 5 page 136-140

-Table 1. and Shannon (Figure 3B): This sentence is not understood.

This sentence has been modified (line 128).

-Is it necessary to express the P-value between WE and KOE in this sentence?

The P-value has been eliminated, since did not provide any relevant information.

- There was a trend, but no statistical difference was observed.

Thank you, we have modified the sentence to indicate that it was a trend, but no significative, line 130, “These results show a trend in alcohol treated animals to have a lower bacterial diversity than control animals”.

 -Figure 6: It is necessary to increase readability by expressing the dominant group according to the direction.

Figure 6 has been modified to make it more visual and easier to understand.

Reviewer 2 Report

In this paper, authors well described the pivotal role of TLR-4 receptor in the development and progression of the disease caused by alcohol consumption abuse. Moreover, they showed the correlation of the lack of the gene of such receptor with a decreased (almost lack) intensity of the intestinal inflammatory response. The topic is important and has a novelty in the point of the continuous growing awareness of the efficacy of novel therapeutic approaches against alcohol-derived diseases or dysbiosis-derived diseases.

However, before publication, the following corrections are mandatory:

- line 33: “microbiota is…”

- line 35-36: sentence should be revised…

- line 42: I would start by saying: “However, an alteration....”

- line 45: please delete “and”

- throughout the text phyla should be italicized….

- line 238: “epithelial ion channel) WAS substantially reduced…”

- line 396-397: please provide the Ethical Committee approval number

- reference style should be deeply revised

Author Response

  • line 33: “microbiota is…”
  • line 35-36: sentence should be revised…
  • line 42: I would start by saying: “However, an alteration....”
  • line 45: please delete “and”
  • throughout the text phyla should be italicized….
  • line 238: “epithelial ion channel) WAS substantially reduced…”
  • line 396-397: please provide the Ethical Committee approval number
  • reference style should be deeply revised

All the typos and corrections suggested by the reviewer have been modified.

Round 2

Reviewer 1 Report

Although the causal relationship, which was a major limitation, was not resolved, results on the relationships between TLR4 and alcohol and gut microbiota are informative. 

Content that did not prove a causal relationship should not be included in the abstract and conclusion. 

  • 1page 18~20:  It is better to change the hypothesis of Abstract to the revised content.
  • 19page 482-484: This conclusion needs to be revised as it has not been directly proved whether the decrease in inflammation caused by alcohol in TLR4 K/O mice is due to changes in gut microbiota.

Author Response

Although the causal relationship, which was a major limitation, was not resolved, results on the relationships between TLR4 and alcohol and gut microbiota are informative.

Content that did not prove a causal relationship should not be included in the abstract and conclusion.

1page 18~20: It is better to change the hypothesis of Abstract to the revised content.

As suggested, the hypothesis of the abstract has been changed.

19page 482-484: This conclusion needs to be revised as it has not been directly proved whether the decrease in inflammation caused by alcohol in TLR4 K/O mice is due to changes in gut microbiota.

As recommended, we have now altered the above sentence in the conclusions.